# Electrochemical Corrosion Resistance of Ni and Co Bonded Near-Nano and Nanostructured Cemented Carbides [†]

**Tamara Aleksandrov Fabijanić [1,*]**, **Marin Kurtela [1]**, **Irbas Škrinjarić [1]**, **Johannes Pötschke [2]** and **Markus Mayer [2]**

[1] Faculty of Mechanical Engineering and Naval Architecture, University of Zagreb, Ivana Lučića 5, 10000 Zagreb, Croatia; marin.kurtela@fsb.hr (M.K.); irbas.skrinjaric@fsb.hr (I.Š.)

[2] Fraunhofer IKTS, Fraunhofer Institute for Ceramic Technologies and Systems, Winterbergstrasse 28, 01277 Dresden, Germany; johannes.poetschke@ikts.fraunhofer.de (J.P.); markus.mayer@ikts.fraunhofer.de (M.M.)

[*] Correspondence: amara.aleksandrov@fsb.hr; Tel.: +385-1611-8389

[†] This paper is an extended version of paper Comparison of nickel and cobalt bonded nanoscaled hardmetals published in Euro PM 2019, Maastricht, Netherlands, 13–17 October 2019.

**Abstract:** The advantages of nanostructured cemented carbides are a uniform, homogenous microstructure and superior, high uniform mechanical properties, which makes them the best choice for wear-resistant applications. Wear-resistant applications in the chemical and petroleum industry, besides mechanical properties, require corrosion resistance of the parts. Co as a binder is not an optimal solution due to selective dissolution in an acidic environment. Thus, the development of cemented carbides with alternative binders to increase the corrosion resistance but still retaining mechanical properties is of common interest. Starting mixtures with WC powder, grain growth inhibitors GGIs; VC and $Cr_3C_2$, and an identical binder amount of 11-wt.% were prepared. GGIs were added to retain the size of the starting WC powder in the sintered samples. The parameters of the powder metallurgy process were adapted, and samples have been successfully consolidated. A very fine homogeneous microstructure with relatively uniform grain-size distribution and without microstructural defects in the form of carbide agglomerates and abnormal grain growth was achieved for both Ni-bonded and Co-bonded samples. Achieved mechanical properties, Vickers hardness, and Palmqvist toughness, of Ni-bonded near-nanostructured cemented carbides are slightly lower but still comparable to Co-bonded nanostructured cemented carbides. Two samples of each grade were researched by different electrochemical direct current corrosion techniques. The open circuit potential $E_{corr}$, the linear polarisation resistance (LPR), the Tafel extrapolation method, and the electrochemical impedance spectroscopy (EIS) at room temperature in the solution of 3.5% NaCl. From the carried research, it was found that chemical composition of the binder significantly influenced the electrochemical corrosion resistance. Better corrosion resistance was observed for Ni-bonded samples compared to Co-bonded samples. The corrosion rate of Ni-bonded cemented carbides is approximately four times lower compared to Co-bonded cemented carbides.

**Keywords:** near-nano and nanostructured cemented carbides; Co; Ni; electrochemical corrosion resistance

## 1. Introduction

Despite the constant search for adequate alternative binders, Co-based cemented carbides are still widely used because of their unique set of properties such as high hardness, toughness, and

thermal shock resistance [1]. Although Co as a binder presents good mechanical properties, superior wetting ability, and beneficial WC solubility, it has environmental pollution, various toxicity issues, and, ultimately, high cost due to increased demand in the electric vehicle industry [1–4]. On the other hand, Ni and Fe are two binders regularly implied as to the best alternative to Co due to their low toxicity, moderately low price, and good wettability of WC [5]. Besides a lower price, Ni-bonded cemented carbides show superior resistance to oxidation and to a wide variety of other corrosive media compared to Co-bonded hard metals [5–7]. Although it is not a prime requirement of cemented carbides, corrosion resistance is important in various applications of cemented carbides especially in the chemical and petroleum industry where the parts are exposed to an aggressive environment [8,9]. Due to excellent oxidation and corrosion resistance, Ni-based cemented carbides can find an advantage in several applications such as wear-resistant parts for pumps, cutting tools in contact with emulsions, elements subjected to organic compounds, or choke valves [10–15]. For example, choke valves are exposed to a highly erosive and corrosive environment where the acid is transported back through the whole production system with the production stream [11–15]. The effects during abrasion or erosion processes are enhanced by electrochemical corrosion and, therefore, corrosion resistance is meaningful criteria in the selection of cemented carbide products in oil and gas production or a similar environment [5,10,11]. However, it is also important for alternative binders to feature good mechanical properties like hardness and wear resistance even at an elevated temperature for various kinds of tools [10].

The main reason why nickel is the first choice in replacing cobalt as an alternative binder is due to a similar crystal structure to cobalt. The face centered cubic FCC cobalt has a lattice parameter of 0.354 nm, which is slightly above an FCC nickel lattice parameter of 0.352 nm [2]. Cobalt has higher hardening rates because of the lower stacking fault energy, which is the opposite to Ni and, thus, WC-Ni composites have inferior hardness and strength to those exhibited by WC-Co [11]. Poorer mechanical properties may be compensated by reducing the grain of the carbide phase. Accordingly, Ni-bonded cemented carbides with ultra-fine or nano-scaled WC grains can also have good surface roughness with a mirror appearance, high hardness, and strength [1,4]. Thus, production of ductile and dense composites while having similar mechanical properties as common WC-Co grades can be achieved with the Ni binder and near nano and nano WC powders [2,12].

In the literature, most of the research is focused on conventional cemented carbides with grain sizes bigger than 200 nm with Co and alternative binders such as Ni and FeNi. The aim of the research was to develop nanostructured cemented carbides with different binders and to compare the properties of developed grades with a focus on the electrochemical corrosion resistance.

## 2. Materials and Methods

Two starting mixtures with WC powder WC DN 2.5 (H.C. Starck GmbH, Glosar, Germany) and identical binder amount of 11 wt.% were prepared in this research. Co (Umicore, Bruxelles, Belgium) and Ni (Eurotungsten, Grenoble, France) were used as binders in the starting mixtures. VC and $Cr_3C_2$ (both H.C: Starck, GmbH, Glosar, Germany) were added as grain growth inhibitors (GGIs) to both starting mixtures in equal quantity. One of the biggest problems of sintering near nano and nano-scaled powders is the preservation of small average grain sizes in the sintered product [16,17]. VC is considered to be the most influential GGI in WC-Ni cemented carbides followed by TaC, $Cr_3C_2$, TiC, and ZrC [11]. The addition of $Cr_3C_2$ has less influence on the WC-Ni grain refinement and hardness in comparison with VC cemented carbide. Both VC and $Cr_3C_2$ reduce the grain growth of the WC hard phase and have a high solubility in the binder [11,18]. An addition of the refractory metal carbides, VC, and $Cr_3C_2$ in the starting mixtures was relatively high to retain the size of the starting powder in the sintered samples.

The powder metallurgy process consisted of mixing in a horizontal ball mill (Zoz GmbH, Wenden, Germany) for 24 h in n-heptane as a milling agent, compacting to bars at 200 MPa by uniaxial die pressing type CA-NCII 250 (Osterwalder AG, Lyss, Switzerland) at room temperature, dewaxing, and

the sinter- hot isostatic pressing HIP process by furnace FPW280/600-3-2200-100 PS (FCT Anlagenbau GmbH, Sonneberg, Germany) in one cycle at 1350 °C for Co-bonded cemented carbides and 1450 °C for Ni-bonded cemented carbides. Overall, two samples from each mixture were selected for the research to ensure the repeatability of the results. The characteristics of the starting mixtures and samples are presented in Table 1.

**Table 1.** Starting powders and sample characteristics.

| Mixture | Sample | Starting Powders | Grain Size $d_{BET}$, nm | Specific Surface, $m^2$/g | Content, wt.% |
|---------|--------|------------------|--------------------------|---------------------------|----------------|
| WC-11Ni | WC-11Ni-a WC-11Ni-b | WC DN 2.5 (H. C. Starck) Ni 2800B (Eurotungsten) $Cr_3C_2$ 160 (H. C. Starck) VC 160 (H. C. Starck) | 150 720 450 350 | 2.6 0.9 2.0 3.0 | remaining 11.0 0.935 0.605 |
| WC-11Co | WC-11Co-c WC-11Co-d | WC DN 2.5 (H. C. Starck) HMP Co (Umicore) $Cr_3C_2$ 160 (H. C. Starck) VC 160 (H. C. Starck) | 150 240 450 350 | 2.6 2.8 2.0 3.0 | remaining 11 0.935 0.605 |

After sintering, the characterization of samples was performed. Measurements of density, magnetic properties, saturation moment, and coercivity were performed. Density measurement was performed according to ISO 3369. Measured density was compared to theoretical density, which was calculated by the rule of mixture for full binder saturation. Coercive force by Förster Koerzimat 1.096, according to ISO 3326 and magnetic saturation measurement by Setaram, and Sigmameter according to D6025 were measured for WC-Co samples. The sample's surface was prepared by grinding and polishing for the purpose of porosity measurement and microstructure analysis. Grinding was performed with a diamond disc MD-Piano 120 (Struers, Ballerup, Copenhagen), which was followed by polishing with diamond pastes DP Piano (Struers) with particle sizes of 15, 9, and 3 μm. Etching for 2–3 s in a Murakami reagent was performed for the purpose of checking the $\eta$-phase [19]. Porosity analysis and detailed microstructural analysis was performed by optical microscopy (Olympus, Shinjuku City, Tokio, Japan) and a field emission scanning electron microscope FESEM (Zeiss, Oberkochen, Germany). The degree of porosity and uncombined carbon was determined by the comparison of polished surfaces with photo micrographs from the standard ISO 4505:1978. The Palmqvist toughness test was used for determining Vickers hardness HV30 and fracture Palmqvist toughness. Palmqvist toughness, $W_k$, was determined by measuring the total length of cracks emanating from four corners of Vickers hardness indentation using the load of 294 N according to ISO 28079:2009 at room temperature. Five indentations were made on WC-11Ni-a and WC-11Co-c samples.

For measuring the electrochemical corrosion resistance, two samples of each mixture were researched by direct current DC and alternating current AC techniques at room temperature (20 ± 2). The surface of samples was polished and ultrasonically cleaned for the purpose of the electrochemical measurement. The polished sample surface with the exposed area of 1 $cm^2$ was immersed into the cell with electrolyte solution of 3.5% NaCl with pH = 6.6. Saturated calomel electrode SCE was used as a reference electrode while graphite wires were used as a counter electrode. Haber-Luggin capillary was inserted close to the working electrode to reduce the influence of Ohmic drop. Different electrochemical direct current corrosion techniques with the following order were used. The open cuircit potential $E_{corr}$, the linear polarisation resistance (LPR), and the Tafel extrapolation method were applied. Open circuit potential ($E_{corr}$) versus SCE was monitored for 30 min prior to polarization. LPR in the potential range from -0.02 V vs. open circuit potential to 0.02 V vs. open circuit potential with a potential scan rate of 0.167 mV/s was performed. Tafel extrapolation was performed in the potential range from −0.25 V vs. open circuit potential to 0.25 V vs. open circuit potential, total points 1001 with the scan rate of 0.167 mV/s to estimate the Co and Ni effect on corrosion resistance and the anodic and cathodic partial electrode reactions. After DC techniques, electrochemical impedance spectroscopy (EIS) from

AC techniques was carried out to investigate the corrosion process at the interface between the sample surface and the electrolyte solution. The potential frequency applied in the research ranged from 100 kHz (start frequency) to 0.001 Hz (end frequency) with an amplitude of 10 mV root mean square RMS, number of points was 70, and points per decade were 10. The results were analyzed by suitable electrical equivalent circuit (EEC) and presented in Nyquist and Bode plots. The measurements were performed on the potentiostat AMETEK, Princeton applied research, model VersaSTAT3 with a typical three-electrode cell set up at room temperature in relation to the reference saturated calomel electrode (SCE) with known potential of +0.242 V, according to standard hydrogen electrode. The corrosion parameters were recorded graphically and analytically using the software SoftCorr III (AMETEK Scientific Instruments, Princeton applied research, Berwyn, PA, USA).

## 3. Results

### 3.1. Characteristics of Ni-Bonded and Co-Bonded Near-Nanostructured Cemented Carbides

The measured density of WC-11Co samples amounts to 14.13 g/cm$^3$, which is about 99.8% relative density. The magnetic saturation of 16.6 μTm$^3$/kg is about 81% of the theoretical magnetic saturation. Relative magnetic saturation with values between 75% and 95% indicates that the two-phase WC-Co microstructure was obtained where no η-phase and no free carbon is expected, which was also confirmed by optical microscopy and FESEM. Measured coercive force amounts to 41.1 kA/m, which indicates that consolidated WC-11 Co samples are in the nano range.

The measured density of the WC–11 Ni amounts to 14.28 g/cm$^3$, which is about 100.8% relative density. Higher density might be related to a high alloyed Ni binder and a possible η-phase. Ni-bonded cemented carbides have an optimal carbon window at smaller carbon contents compared to Co cemented carbides, which is the reason why Ni binders are mostly highly alloyed. This leads to a higher density. The magnetic saturation measurement is not possible for WC-11 Ni samples due to not ferromagnetic properties of the Ni binder after sintering.

The analysis of the polished and Murakami etched surface showed no η-phase or free carbon for WC-11 Ni and WC-11 Co samples. The porosity of samples is estimated to A02, B00, and C00. Overall, a good cemented carbide quality was achieved for both grades. The microstructure of WC-11 Ni and WC-11 Co samples are presented in Figure 1.

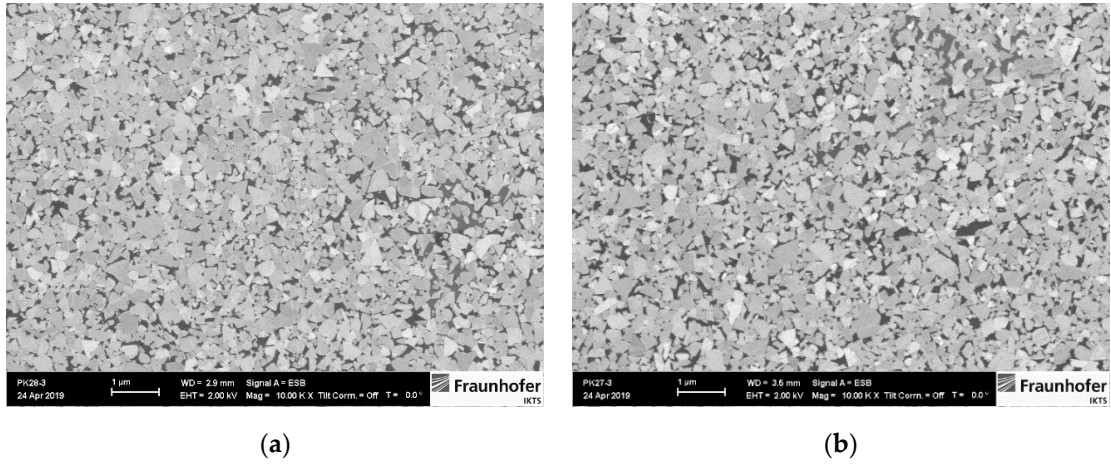

(**a**)      (**b**)

**Figure 1.** Microstructure of samples (**a**) WC-11 Ni-a and (**b**) WC-11 Co-c.

Mean WC grain size of WC-11 Ni sample determined by the linear intercept method is around 250 nm. Therefore, the developed WC-11 Ni cemented carbide can be classified as near-nanostructured cemented carbide ($d_{WC}$ > 200 nm). Even though, an addition of the refractory metal carbides, VC, and Cr$_3$C$_2$ in the starting mixtures was relatively high, to retain the size of the starting powder in

the WC-11 Ni sintered samples and to achieve a real nano structure, the optimization of the mixture and process parameters are required. Mean hardness value of 1810 HV30 and $W_k$ of 8.51 MN/m$^{3/2}$ was measured for the WC-11 Ni-a sample. Slightly higher values of Vickers hardness and Palmqvist fracture toughness with the mean hardness value of 1879 HV30 and $W_k$ of 9.35 MN/m$^{3/2}$ was measured for the WC-11 Co-c sample.

### 3.2. Results of Electrochemical DC Measurements

The results of electrochemical DC techniques, open circuit potential $E_{corr}$, linear polarisation resistance LPR, and potentio-dynamic polarization curves (Tafel) are presented in Table 2.

**Table 2.** The results of electrochemical DC techniques.

| Sample | $Ts$ [°C] | $E_{corr}$ vs. SCE [mV] | $R_p$ [kΩcm$^2$] | $\beta_a$ [mV/dec] | $\beta_c$ [mV/dec] | $i_{corr}$ [μA/cm$^2$] | $v_{corr}$ [mm/y] |
|---|---|---|---|---|---|---|---|
| WC-11Ni-a | 20 ± 2 | −206 | 7.623 | 88.86 | 88.86 | 0.467 | 0.0127 |
| WC-11Ni-b | 20 ± 2 | −208 | 7.603 | 80.13 | 212.75 | 2.743 | 0.0238 |
| WC-11Co-c | 20 ± 2 | −234 | 5.820 | 80.39 | 305.63 | 5.991 | 0.0540 |
| WC-11Co-d | 20 ± 2 | −284 | 4.954 | 82.80 | 290.08 | 6.149 | 0.0554 |

Where $Ts$—measured temperature, $E_{corr}$—open circuit potential, $R_p$—polarisation resistance, $\beta a$—slope of anodic Tafel curve, $\beta_c$—slope of cathodic Tafel curve, $i_{corr}$—corrosion current density, and $v_{corr}$—corrosion rate.

Open circuit potential (OCP) versus time curves of samples for all samples are presented in Figure 2.

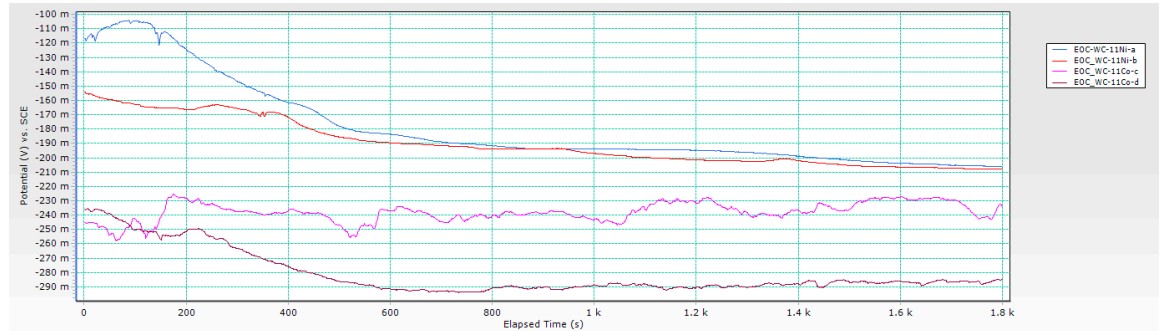

**Figure 2.** Open circuit potential (OCP) versus time curves of samples.

Linear polarization resistance (LPR) curves used to investigate the electrochemical response of samples near their open circuit potential are presented in Figure 3.

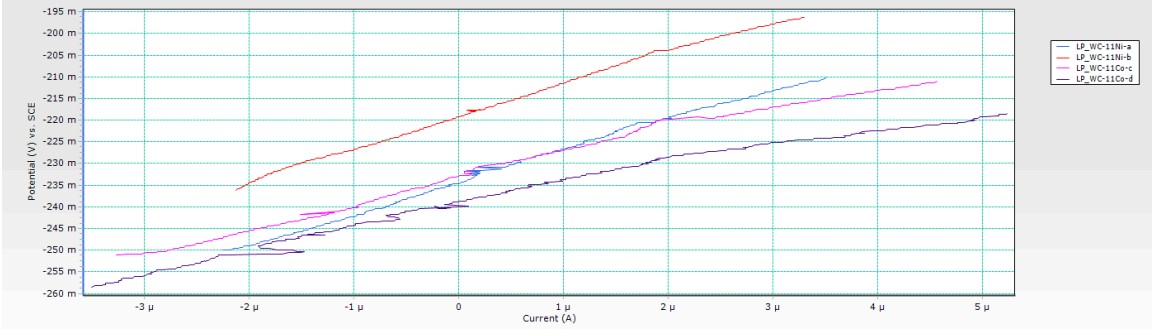

**Figure 3.** Linear polarization curves of samples.

Tafel extrapolation curves for all samples are presented in Figure 4.

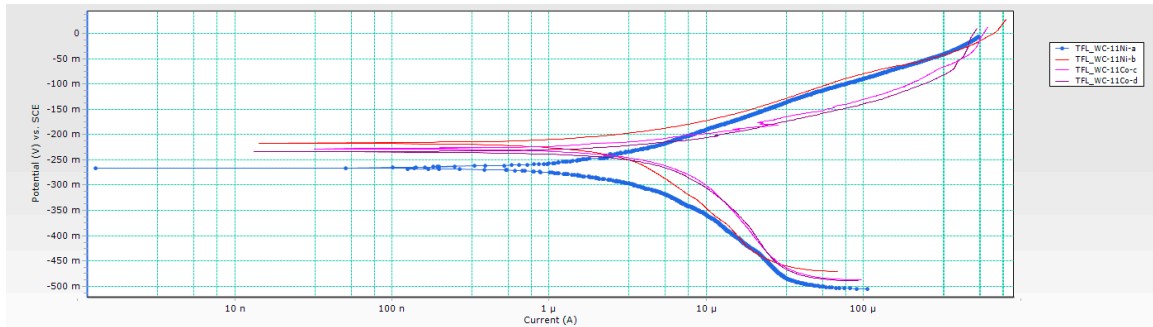

**Figure 4.** Tafel extrapolation curves of samples.

### 3.3. Results of Electrochemical Impedance Spectroscopy (EIS)

The results of electrochemical impedance spectroscopy (EIS) measurements are presented in Table 3.

**Table 3.** The results of electrochemical impedance spectroscopy (EIS) technique.

| Sample | $Ts$ [°C] | $Rs$ [$\Omega$cm$^2$] | $Q$ | $n_1$ | $R_{ct}$ [$\Omega$cm$^2$] |
|---|---|---|---|---|---|
| WC-11Ni-a | 20 ± 2 | 5.201 | 1.047$\cdot\times 10^{-4}$ | 0.935 | 8.888$\cdot\times 10^3$ |
| WC-11Co-c | 20 ± 2 | 5.188 | 1.402$\cdot\times 10^{-4}$ | 0.896 | 6.134$\cdot\times 10^3$ |

Measured and calculated values of the impedance in the Nyquist diagrams were compared. The model of equivalent electrical circuit (EEC) that best describe the reactions and changes on the sample's surface in the electrolyte using data analysis software ZSimpWin Version 3.2 (AMETEK Scientific Instruments, Princeton applied research, Berwyn, PA, USA) was selected. The schematic diagram of the selected EEC circuit is presented in Figure 5.

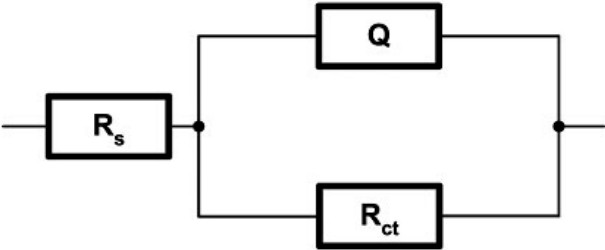

**Figure 5.** Model of an R(QR) electrical circuit.

The same model R(QR) was selected for both Co and Ni bonded cemented carbide samples. The depicted EEC contains the following elements: $R_s$ is a solution resistance between the working electrode and the reference electrode in a three-electrode cell, $R_{ct}$ is polarization resistance or resistance to charge transfer on the electrode/electrolyte interface, and $Q$ is a Constant Phase Element (CPE).

The impedance of a constant phase element is defined by the equation below.

$$Q = \left[Y(j\omega)^n\right]^{-1}, \tag{1}$$

where $Y$ and $n$ ($-1 \leq n \leq 1$) are constants independent on the angular frequency ($\omega$) and temperature. For the value in the range of $0.6 < n \leq 1$, CPE has the physical meaning of capacitance, an ideal inductor for $n = -1$, and an ideal resistor for $n = 0$.

## 4. Discussion

When a metal specimen is in contact with a corrosive medium and the specimen is not connected to any instrumentation, the specimen assumes a potential (relative to a reference electrode) termed the open circuit potential, $E_{corr}$. A specimen at $E_{corr}$ has both anodic and cathodic currents present on its surface [19–22]. However, these currents are exactly equal in magnitude so there is no net current to be measured. The specimen is at equilibrium with the environment. Prior to all electrochemical tests, $E_{corr}$ variation vs. time was monitored for 30 min and the obtained recordings are shown in Table 2. It can be seen that, for measurements in 3.5%NaCl, the potential shifts in the negative direction. It can be seen from Figure 2 that Ni and Co bonded samples showed negative values of $E_{corr}$ in the electrolyte, which indicate the instability on the surfaces and material dissolution [19–24]. It is important to mention that the potential shifts to more negative values may be caused by other effects, e.g., passive film generation and chemical evolution of the surface. As expected, the results show that, under the same chloride solution, temperature, and immersion time, $E_{corr}$ values are more positive for Ni bonded samples compared to Co bonded samples, which indicates better corrosion properties of Ni bonded samples in the test medium [19–24]. Both samples showed the same trend and good repeatability of the results. The polarisation resistance technique was used to measure the polarization resistance $R_p$, which is defined as the resistance of the specimen to oxidation during the application of an external potential [19–22]. $R_p$ was determined by calculating the slope of the linear region of the plots and can be used to assess the relative ability of a material to resist corrosion. Since all samples were of equal surface area, the sample with the highest $R_p$ and, thus, the lowest $i_{corr}$ has the highest corrosion resistance. It can be seen from Figure 3 that the curves are not ideally linear. More linear curves were registered for WC-Ni samples. In addition, it can be observed for both Ni-bonded samples that the slope of the curves is steeper compared to WC-Co samples, which are related to higher polarization resistance values. Tafel plots are used to measure the corrosion current $i_{corr}$, so that the corrosion rate can be calculated. Tafel plot yields icorr directly or it yields the Tafel constants $\beta_A$ and $\beta_C$ [19–22]. Tafel extrapolation curves of researched samples showed anodic behaviour in the solution of 3.5% NaCl with pH = 6.6, which might relate to the oxidation reaction of the samples and the atom passes into the electrolyte as a cation. The slope of the anodic Tafel curve $\beta_A$ showed no general trend and is around 80 mV/dec, while the slope of the cathodic Tafel curve $\beta_C$ is higher for the Co-bonded cemented carbides.

Both samples showed relatively good repeatability. Slightly better match of the Tafel curves was noted for WC-Co samples.

Higher values of $R_p$ and lower values of $i_{corr}$ were measured for WC-Ni samples. The highest $R_p$ of 7.623 kΩcm$^2$ and the lowest corrosion current density $i_{corr}$ of 0.467 μA/cm$^2$ was measured for sample WC-11Ni-a. The highest value of $i_{corr}$ of 6.149 μA/cm$^2$ and the lowest $R_p$ of 4.954 kΩcm$^2$ was measured for sample WC-11Co-d. The relationship between the electrochemical parameters, polarization resistance $R_p$, and corrosion current density icorr with respect to binder composition is presented graphically in Figure 6.

The corrosion rate $v_{corr}$ of Ni-bonded cemented carbides is approximately four times lower when compared to Co-bonded cemented carbides. The lowest $v_{corr}$ of 0.0127 mm/y was measured for WC-11 Ni-a while the highest of 0.0554 mm/y was measured for WC-11Co-d. Such a behaviour is related to excellent oxidation and corrosion resistance of the Ni binder. Bigger deviation was noted for WC-Ni samples.

The Nyquist and Bode diagrams of samples with the corresponding EEC model applied for the simulation of EIS results are presented in Figures 7–10.

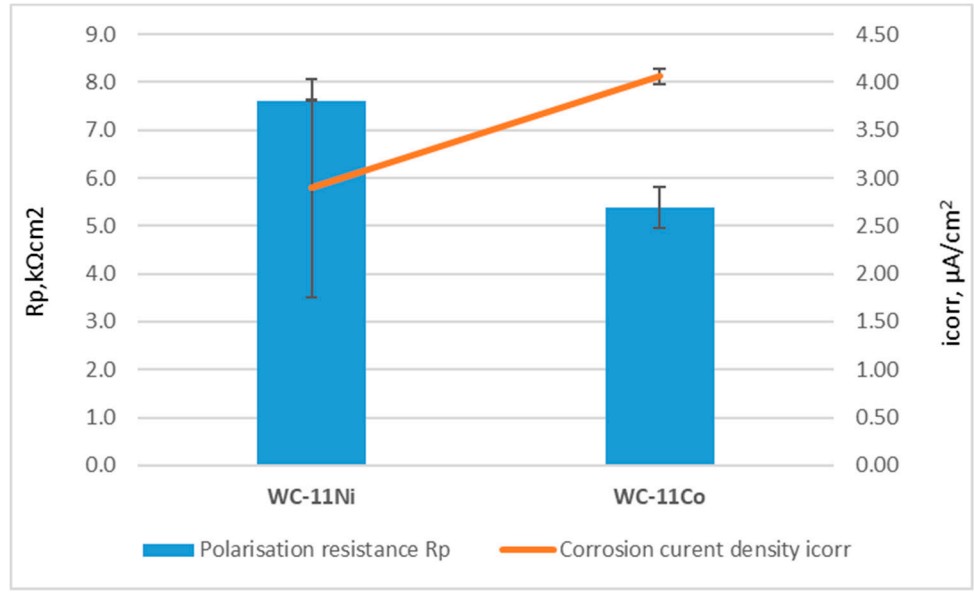

**Figure 6.** The dependence of $R_p$ and $i_{corr}$ in relation to binder composition.

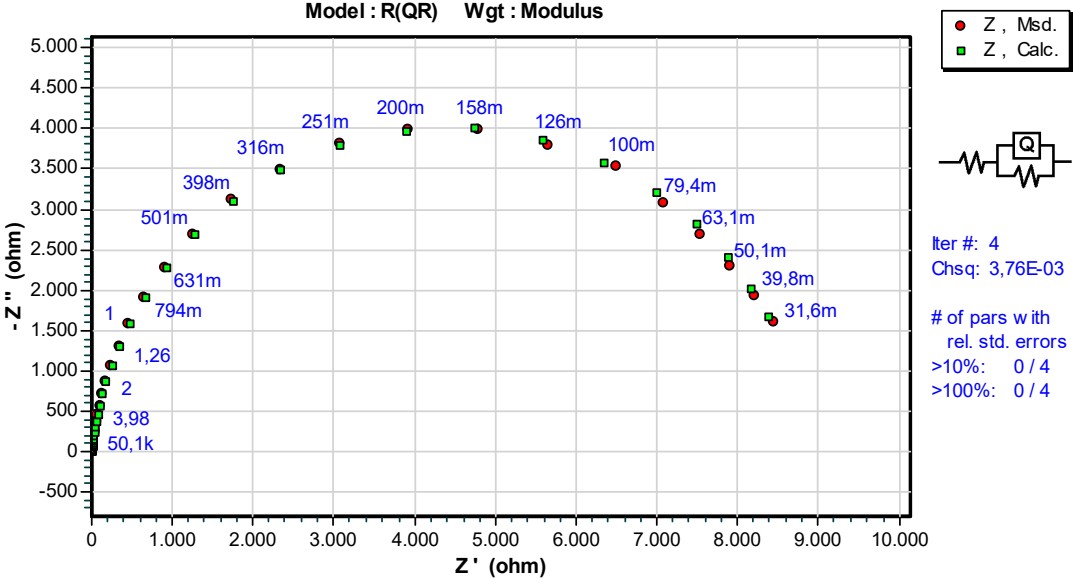

**Figure 7.** Nyquist plot of WC-11 Ni-a.

EIS results are in line with the results obtained by DC linear polarization techniques. Higher $R_{ct}$ of 8.888 kΩcm² was measured for WC-11 Ni-a sample compared to WC-11 Co-c sample for which $R_{ct}$ amounts to 6.134 kΩcm² confirming better corrosion resistance of Ni-bonded cemented carbides as presented in Table 3.

The radius r of the capacitive semi-circles in the Nyquist diagrams presented in Figures 7 and 9 differ for the WC-11 Ni-a and WC-11 Co-c samples. The diameter of the capacitive loop of Ni bonded sample is larger compared to the Co-bonded sample. Subsequently, the decrease in diameter of the capacitance loop is related to the weaker protective ability of the surface film.

In previous research, the corrosion resistance of nanostructured WC-Co cemented carbides with different Co content in the range from 5 wt.% to 15 wt.% in the same solution was investigated [23,24]. Two-phase WC-5Co cemented carbide showed lower $R_p$ of 1.324 kΩcm² and higher $i_{corr}$ of 35.32 μA/cm² compared to Co and Ni-bonded cemented carbides consolidated and researched in this paper [23,24]. In addition, two-phase WC-10Co cemented carbides showed lower $R_p$ of 0.963 kΩcm²

and higher $i_{corr}$ of 34.80 μA/cm². Both cemented carbides with lower Co content from previous research were consolidated from the same starting WC powder and the same Co type, but the content of GGIs added to the starting mixture was lower, which indicates that GGIs influenced the results. It is important to mention that besides GGIs, C content added to the starting mixtures differs, which is found to significantly influence the electrochemical corrosion resistance. Better corrosion resistance was observed for samples with lower content of C added and lower magnetic saturation [23,24]. Considering all differences between the compared cemented carbides, it can be concluded that the chemical composition of the binder and microstructural characteristics might have a more significant influence than the weight content of the binder phase. Future research will focus on the influence of GGIs and other alternative binders on the electrochemical corrosion resistance of nanostructured cemented carbides.

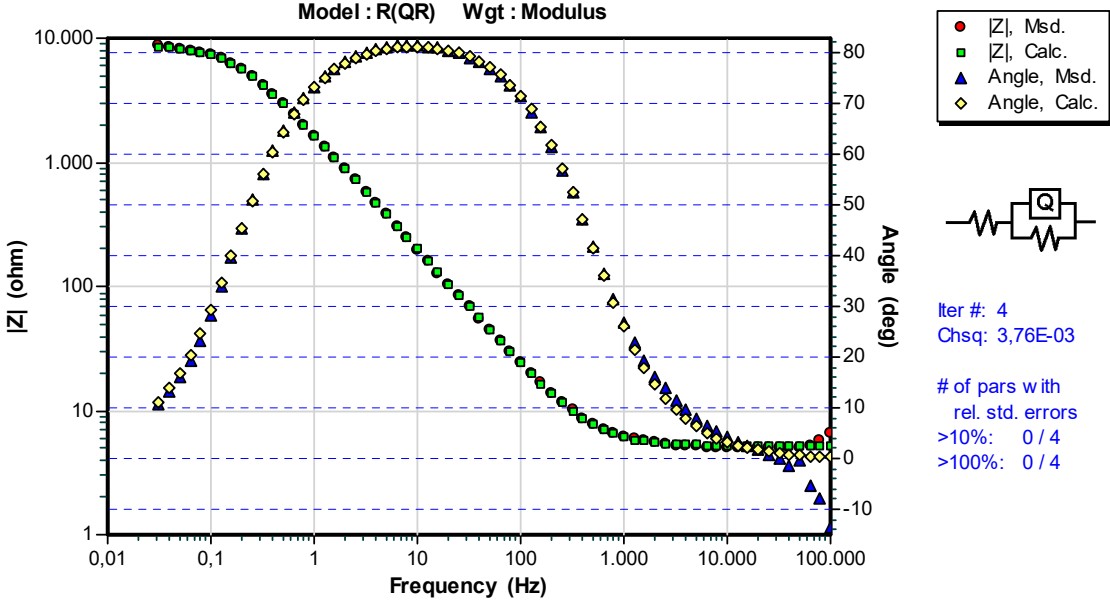

**Figure 8.** Bode plot of WC-11 Ni-a.

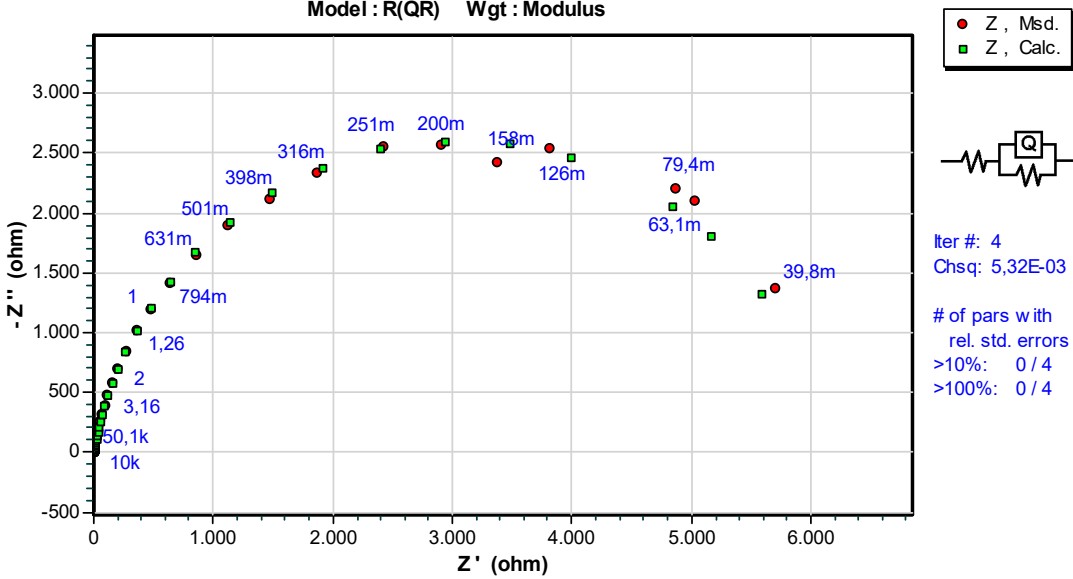

**Figure 9.** Nyquist plot of WC-11 Co-c.

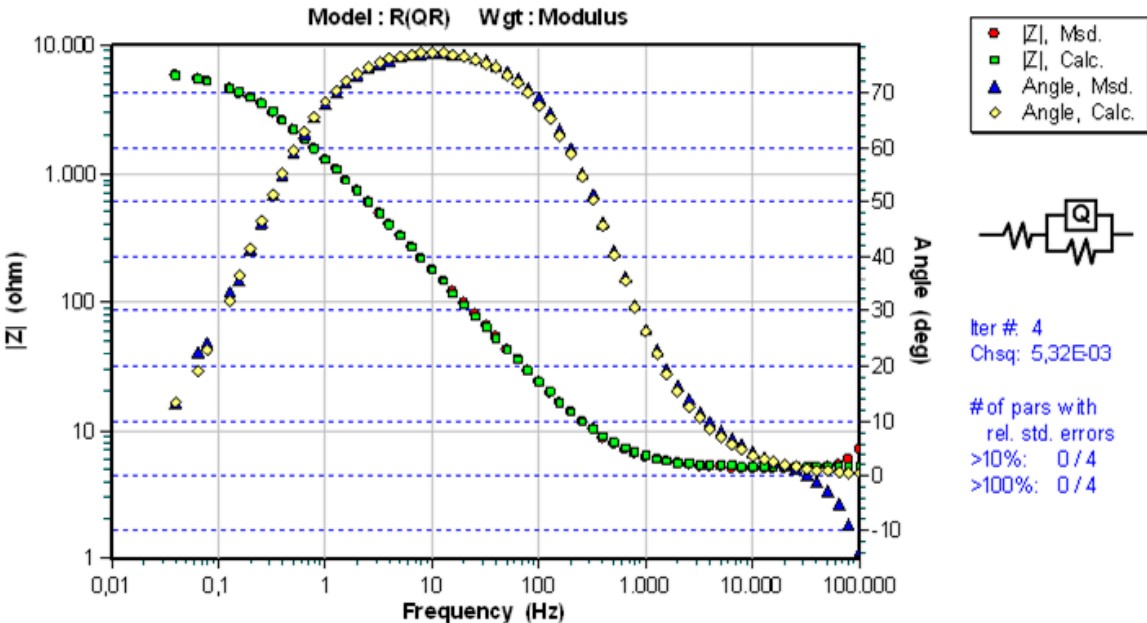

**Figure 10.** Bode plot of WC-11 Co-c.

## 5. Conclusions

The following conclusions can be drawn from the conducted research.

i. The samples of Ni and Co-bonded near-nanostructured and nanostructured cemented carbides have been successfully consolidated. A very fine homogeneous microstructure with relatively uniform grain-size distribution, and without microstructural defects in the form of carbide agglomerates and abnormal grain growth, was achieved for both Ni and Co-bonded samples. Achieved mechanical properties, Vickers hardness, and Palmqvist toughness of Ni-bonded near-nanostructured cemented carbides are slightly lower but still comparable with Co-bonded nanostructured cemented carbides. Reduction of the WC grain of the starting powder improved the mechanical properties of Ni-bonded cemented carbides.

ii. Chemical composition of the binder significantly influenced the electrochemical corrosion resistance. Better corrosion resistance was observed for Ni-bonded samples compared to Co-bonded samples. Higher values of polarisation resistance $R_\mathrm{p}$ and lower values of corrosion current densities $i_\mathrm{corr}$ were measured for WC-Ni samples. The corrosion rate of Ni-bonded cemented carbides is approximately four times lower compared to Co-bonded cemented carbides.

iii. Chemical composition of the binder and microstructural characteristics might have a more significant influence than the weight content of the binder phase.

**Author Contributions:** M.M. and J.P. consolidated the samples and performed the characterization of samples. T.A.F., I.Š., and M.K. performed the characterization and electrochemical measurements, analyzed the data, and wrote the paper. All authors have read and agreed to the published version of the manuscript.

**Funding:** This work is supported in part by the Croatian Science Foundation under the Project Number UIP-2017-05-6538 Nanostructured hard metals—New challenges for Powder Metallurgy. This work is supported in part by European Regional Development Fund under the Project number KK.01.2.1.01.0079 Research and development of nanostructured hard metals for the development of new products NANO-PRO.

**Conflicts of Interest:** The authors declare no conflict of interest.

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
