# Peer review of "Electrochemical Corrosion Resistance of Ni and Co Bonded Near-Nano and Nanostructured Cemented Carbides†"

_metals, doi:10.3390/met10020224_

Round 1

Reviewer 1 Report

Board comments,

This manuscript studies the influence of the binders (Co and Ni) in the physicochemical properties of the WC such as porosity, magnetic saturation, coercive force and corrosion resistance. The microstructure characterisation of the samples seem to be appropriate but the analyses’ details are unclear in this document. Although the measurements of the magnetic saturation and coercive force appear have been rightly carried out, the experimental setup of these analyses are not detailed in this work. In addition, analysis techniques to evaluate the corrosion resistance of these samples seem to be right but the experimental condition details should be improved in this manuscript.

Although the objectives of the paper appears to have been achieved, some parts of this document should be improved. Thus, it is recommended to consider the following comments regarding the manuscript.

One or two sentences about the issue (the problem to be solved) should be included in the abstract.  

The details of the facility (supplier and model) and experimental conditions used to fabricate these samples should be added to in the part 2. Materials and Methods.

If H.C. Starck, Eurostungsten and HMP are the supplier of the materials, it should be indicated in part 2. Materials and Methods.

The details of facilities (supplier, and model) utilised in optical microscopy, FESEM, XRD, density measurement, porosity analysis and magnetic saturation should be indicated in part 2 Materials and Methods.

Material or type of the counter or auxiliary electrode should be commented in part 2. Materials and Methods.

The experimental conditions of the corrosion potential or open circuit potential should be indicated in part 2. Materials and Methods, e.g. total time data acquisition.

The experimental conditions of the linear polarisation resistance should be indicated in part 2. Materials and Methods, e.g. potential scan rate and range.

The experimental conditions of the Taffel extrapolation or potentiodynamic polarisation curve should be indicated in part 2. Materials and Methods, e.g. potential scan velocity, initial and final potential.

The experimental conditions of the electrochemical impedance spectroscopy (potential amplitude, total points, initial potential and points per decade) should be indicated in part 2. Materials and Methods.

The standard or experimental setup of the polishing process should be detailed in part 2. Materials and Methods.

The standard or experimental setup of the etching dissolution should be commented in part 2. Materials and Methods.

Measured temperature or Ts should be in part 2. Materials and Methods.

Figure 2 could be the graph of the corrosion potential evolution with the time or open circuit potential.

The negative values of the potential in Figure 2 could be due to the reduction of the noble of material. Although this could be due to instability on the surfaces and dissolution, a reference should be included.

The part of 3. Results should be improved with description of the evolution of the potential, analysis of the line polarisation resistance behaviour and comments about branch of the potentiodynamic polarisation curves.

The part of 4. Discussion should be upgraded via the explication of the evaluation of the potential with time, linear polarisation resistance and potentiodynamic polarisation curves. In addition, an explication of the selection of this electrical equivalent circuit should be included in this part.

The references should only be in the discussion part. If you include reference in the experimental setup or results part, it could generate confusions.

Specific comments.

My specific commentaries are the following:

To define acronyms DC and AC 63 line “researched by DC AC techniques at room temperature.

To add reference in 80 line 2saturation (Ref)”.

To define acronym FESEM and XRD 82 line “also confirmed by optical microscopy, FESEM and XRD analysis”.

To include reference in 88 line “to a higher density (REF)”.

96-98 lines “Even though, an addition of refractory metal carbides: VC, Cr3C2 in the starting mixtures was relatively high, to retain the size of the starting powder in the WC-11Ni sintered samples optimization of the mixture and process parameters is required.” Should be in the part 2. Materials and Method.

To replace Rp to Rct in the Table 4.

To replace Figures 6 to Figures 7, 8, 9 and 10 150 line.

To indicate the value of the Rp and icorr 158 line “lower Rp and higher icorr compared to Ni bonded cemented carbides researched in this paper”.

Author Response

Dear reviewer,

thank you for the comments. The manuscript is revised according to yours and board comments. We hope the improved version will be accepted for publishing.

Best regards,

dr.sc. Tamara Aleksandrov Fabijanić

Reviewer 2 Report

Two Nano-structured cemented carbides were fabricated, and their corrosion behavior was studied in a 3.5% NaCl solution. Despite the potential interest of the application of these materials in a corrosive environment, there are several major issues affecting the quality of the manuscript. Therefore, it is not recommended for publication.

I divided my comments into two sections of general and specific comments, as follows.

General comments:

Abstract: What does GGIs stand for? It was introduced later in Materials and Methods section, but it should be introduced here.

Abstract: What does "good material characteristics" mean? Please be specific.

Introduction: The Introduction does not have a proper flow. You may want to consider revising it.

Page 3: Acronyms have been introduced before.

Page 3: “…typically from less than 0,001 Hz…” Please be specific.

Page 3: “…known potential of +0.242 V” Was the potential versus standard hydrogen electrode (SHE)?

Page 4, Table 2: What is the difference between WC-11Ni-a and WC-11Ni-b (also WC-11Co-a and WC-11Co-b)? Why such a relatively large difference was observed for the two samples with similar composition and microstructure?

Critical comments:

The manuscript was poorly written. There are several careless mistakes in the text and graphs. The whole manuscript MUST be revised carefully.

The Materials and Methods section has to be revised clearly explaining the procedure. Please consider the followings:

How about the sample surface area? Did you apply any surface treatment prior to conducting the electrochemical tests? What was the test condition (i.e., potential range) for the potentiodynamic polarization? For how long the corrosion potential was recorded? What was the order of the tests (first Ecorr measurements, then LPR, EIS, potentiodynamic)?

It is not acceptable to show a potential vs. time curve directly from the potentiostat software. Please redraw all figures (except Figures 5 and 6) using proper graphing and data analysis software (e.g., Microsoft Excel).

WHY Figures 7 to 10 were not discussed or explained in the manuscript?

In Discussion section you are supposed to discuss your findings and propose possible mechanisms behind the observed behavior of the materials. However, the Discussion section of this manuscript was simply repeating the Results section.

Author Response

Dear reviewer,

thank you for the comments. The manuscript is revised and improved according to your comments. We hope the revised version will be accepted for publication.

Best regards.

Round 2

Reviewer 1 Report

Board comments,

This document shows an adequate study about the physicochemical properties (density, magnetic saturation, coercive force, hardness and corrosion resistance) of the WC-Co and WC-Ni. The analyses of the corrosion resistances for both materials were carried out via correct number and type of electrochemical analysis techniques, e.g. open circuit potential, linear polarisation resistance, potentiodynamic polarisation curves and electrochemical impedance spectroscopy. Microstructure and porosity of the samples were appropriately evaluated through etching, optical microscopy and scanning electron microscopy. The study of the influence of microstructure in the physicochemical properties of the materials were done adequately.

Although this manuscript has achieved to reach its aims and it can be published, it is recommended to consider the next comments regarding the paper:

Electrochemical analysis technique named as corrosion potential should be called as open circuit potential. This is due to the term of corrosion potential is commonly used to name a thermodynamic property of the corrosion process of materials. Although this electrochemical analysis technique evaluates the evolution of the corrosion potential with the time, it is usually called as open circuit potential.

The details of the grinding and polishing process (e.g. grades of the abrasive papers, types of the diamond pastes or colloidal silica gel) should be included in the 2.Materials and Methods part because this can be of interest to readers.

The evolution of the potential to more negative values with the time because the instability on the surfaces and material dissolution (175 line) should be referenced. This is due to the shift of the potential to more negative values can be caused by other effects, e.g. passive film generation and chemical evolution of the surface.

According to my calculations, number of points should be 70, but you indicate that it is 30 points. My calculations were carried out via multiplication of the decades (7, from 100KHz to 0.01Hz) with points per decade (10). If my calculations are correct, you could modify the value, please.

Specific comments.

My specific commentaries are below:

To include reference in 83 line “seconds in a Murakami reagent (REF)”.

To replace -0.02V by -0.02V vs open circuit potential and 0.02V by 0.02V vs open circuit potential in 102 line “potential range from -0.02V (initial potential) to 0.02V (final potential) with”.

To replace -0.25V by -0.25V vs open circuit potential in 103 line “0.167mV/s was performed. Tafel extrapolation was performed in the potential range form -0.25V”.

To replace 0.25V by 0.25V vs open circuit potential 104 line “(initial potential) to 0.25V (final potential)”.

To add reference in 169-170 lines “A specimen at Ecorr has both anodic and cathodic currents present on its surface (REF)”.

To include reference in 178 line “samples , indicating better corrosion properties of Ni bonded samples in the test medium (REF).”.

To add reference in 181 line “defined as the resistance of the specimen to oxidation during the application of an external potential (REF).”

To include reference in 189 “Tafel plot yields icorr directly or it yields the Tafel constants βa i βc (REF).”

Author Response

Dear Reviewer and Board,

thank you for the constructive comments which improved the quality of the manuscript.  We made changes in the manuscript according to your requirements. Hope you will be satisfied. 

Best regards,

Tamara Aleksandrov Fabijanić

Reviewer 2 Report

I have rejected the first version of this manuscript, due to several critical issues. I will also reject the revised manuscript, as the authors did not address my comments. They are supposed t submit a rebuttal letter describing all changes and address all comments from the reviewers.

Author Response

Dear Reviewer,

I am very sorry to hear that we did not address your comments. We have accepted the comments you gave us in the first round of the reviewing process and corrected the manuscript according to your requests and recommendations. Most of your comments were the same as the board and reviewer 1 comments and they have accepted the changes and manuscript. All changes in the revised version of the manuscript are marked in Track changes. Also, we have submitted a letter describing all the changes as a separate file in the first round of the reviewing process. The Metals template was used to submit answers to reviewers. 

I submit the letter with the answers from the first round of the reviewing process again, maybe something went wrong during the submission process.

We apologize for any inconvenience.

Best regards,

Tamara Aleksandrov Fabijanić
